# Global Health Perspectives on Race in Research: Neocolonial Extraction and Local Marginalization

**DOI:** 10.3390/ijerph20136210

**Published:** 2023-06-23

**Authors:** Akhenaten Siankam Tankwanchi, Emmanuella N. Asabor, Sten H. Vermund

**Affiliations:** 1Department of Health Systems and Population Health, University of Washington School of Public Health, Seattle, WA 98195, USA; 2Department of Epidemiology of Microbial Diseases, Yale School of Public Health, New Haven, CT 06510, USA; emmanuella.asabor@yale.edu (E.N.A.); sten.vermund@yale.edu (S.H.V.); 3Department of Pediatrics, Yale School of Medicine, New Haven, CT 06510, USA

**Keywords:** global health, human subjects research, international health, medical racism, research ethics, research neocolonialism, tropical medicine

## Abstract

Best practices in global health training prioritize leadership and engagement from investigators from low- and middle-income countries (LMICs), along with conscientious community consultation and research that benefits local participants and autochthonous communities. However, well into the 20th century, international research and clinical care remain rife with paternalism, extractive practices, and racist ideation, with race presumed to explain vulnerability or protection from various diseases, despite scientific evidence for far more precise mechanisms for infectious disease. We highlight experiences in global research on health and illness among indigenous populations in LMICs, seeking to clarify what is both scientifically essential and ethically desirable in research with human subjects; we apply a critical view towards race and racism as historically distorting elements that must be acknowledged and overcome.

## 1. Introduction

The growing literature from non-Western authors in *Index Medicus* suggests a recent shift in human subjects research from high-income nations to low- and middle-income countries (LMICs). As of December 2022, searches of the PubMed database of the National Library of Medicine yielded significant increases in citations from the past decade for Africa (50%), India (67%), and China (81.5%), versus 41% for the United States (USA), these being just a few illustrative comparators (Table 1). While there are still vast disparities between Africa, India, China, and the USA, one can nonetheless assume the continued migration of research focus from high-income nations to LMICs, where the populations being studied have been historically underrepresented in global human subjects research, and when included in such research have too often received little to no benefit from the research or its findings [1].

As reviewed in this article, an updated and expanded version of a chapter from the forthcoming APHA book *Race and Research* [2], the lived experience of persons in LMICs has become a more central focus for researchers and research funders in recent years, whereas colonial-era research work was too often concerned with studying the bodies and physical health of local denizens, producing tropical disease research that might principally benefit White foreigners with ample resources, e.g., colonists, military personnel, tourists, and business travelers [3]. By the late 1990s, descriptive and interventional research that sought to understand the full human experience of human disease began to expand in LMICs where research was already established, e.g., South Africa, Kenya, Zimbabwe, Thailand, Malaysia, China, Chile, Brazil, and Mexico. Expansion into nations whose prior research experience was limited, as with Zambia and Liberia, was accelerated by such global threats as HIV/AIDS and Ebola virus, respectively [4,5,6,7,8,9,10]. Hence, an increasing proportion of international studies involved diverse persons of color, and local investigators were engaged in and trained for leadership in such studies. Today, these projects increasingly include research on non-communicable diseases (NCDs), including mental health conditions. While NCDs in LMICs may not appear as threatening to high-income nations’ economies and overall health as are pandemic threats, they nonetheless represent hazards to global markets through their effects on consumption, trade, and workforces.

One can view this transition from “tropical medicine” to “global health” as expanding the work’s relevance to LMICs, independent of its applicability to high-income nations. “Tropical medicine” research often involved researchers from Europe and North America conducting studies in Africa, Asia, and South America that were motivated by interests of the colonial inhabitants from high-income countries; the needs and concerns of the peoples being studied may have been influential in study design, but were too often neglected in implementing the findings of the research. “Global health” research in its ideal form is more expansive, focused on both prevention and clinical care, addressing the problems of the persons being studied, and increasingly conducted by investigators from the countries where the research is promulgated. In addressing experiences of research on health and illness in LMICs, particularly among Indigenous populations, we seek to clarify elements of what is both scientifically essential and ethically desirable in human subjects research. We present challenges and opportunities for ensuring the ethical conduct of research in LMICs, with a special focus on race and racism as historically distorting elements that must be overcome.

## 2. Research Neocolonialism

As early as the 15th century and as recently as the 21st century, European colonial powers and later American imperial forces were engaged in extracting natural resources, slaves, low-cost labor (often indentured), and/or geopolitically strategic land through their colonies and territories [11,12,13,14,15]. The field of tropical medicine was oriented towards the prevention, diagnosis, and treatment of diseases that might afflict expatriates or reduce the efficiency of a local workforce. Of note, this remains true under globalized capitalism, albeit tempered by more modern and enlightened views of donor nations’ research funding agencies and the influence of private foundations such as Gates, Rockefeller, Ford, and the Wellcome Trust. The evolution of participatory research better engages the community than “parachute research”, in which researchers from high-income nations gathered data and specimens from “volunteers” who may have given just rudimentary “informed consent”, returning the data and specimens to the high-income country for analysis and academic publication [16]. Particularly when interventions were costlier, the results of tropical medicine research were less likely to benefit the communities that made the research possible, since discoveries were not translated back to those same communities.

Race was a persistent theme for colonial researchers. Some believed in paternalism, that they could decide what would be best for African, Asian, Caribbean, Latin American, and Pacific Islander populations. In the late 19th and early 20th centuries, for prominent medical, public health, and political authorities, “it was difficult, if not impossible, to render coherent the masses of anecdotal observations on the racial distribution of disease… nevertheless, most explanations (and speculations) continued to presume some racial essence, or constitution, that was more or less resistant to some diseases, and liable to others” [17] (p. 101). Race was seen to explain such special immunity, as well as unique susceptibility. For example, a French biologist wrote in 1883 that “negritoid-polynesian” [17] (p. 100) races were immune to scarlet fever. Special immunity on the basis of race was used to justify medical neglect. In contrast, unique susceptibilities on the basis of race cast non-White persons as contagious and inferior, justifying racial separation and reifying the racist hierarchy that supported the colonial/imperial enterprise. Even when physicians and scientists acknowledged the incomplete and at times incoherent nature of racial explanations for disease distribution, these explanations still had currency due to prevailing notions of an inherent racial inferiority of people of color. 

While incidence rates differed across races, many possible explanations could have been marshalled in 1883 to explain this difference [17]. Edward Jenner began developing a method of smallpox inoculation in the late 18th century, and by the early 19th century, smallpox vaccination was common in Europe and North America [18]. In the 1870s, Joseph Lister developed aseptic surgical techniques and other practical applications of germ theory [19]. Louis Pasteur communicated his work on weakened *Pasteurella* and anthrax for the purpose of inoculation in 1880 and 1881 [20,21]. Robert Koch published his identification of the causative organism of tuberculosis in 1882 [22]. Yet, well into the 20th century, scientific authorities continued to reify biological racial difference despite evidence for more precise mechanisms for infectious disease.

A 1920 presidential address to the British Medical Association used race to explain the distribution of malaria: “The constitution of the Negro is more tolerant of malaria than that of the Caucasian race” [17] (p. 100). (We comment below on sickle cell trait.) In 1903, in the first edition of *Infection and Immunity*, retired US Army Surgeon-General George M. Sternberg said of malaria and yellow fever that “the negro is less susceptible to yellow fever and the malarial fevers than the white race; on the other hand, smallpox is exceptionally fatal among negroes and dark-skinned races” [23] (p. 24). Biological difference on the basis of race was assumed, and acquired immunity was discounted. 

Complicating a discourse of race and research in LMICs is an observation that there are some examples in which genetic characteristics that are geographically clustered do influence risk of infection and/or disease. One example is malaria, a major contributor to infant mortality in hyperendemic areas of sub-Saharan Africa with residual partial acquired immunity among survivors. Lower rates of disease and higher rates of survival with the *Plasmodium falciparum* form of malaria are noted among persons with the sickle cell trait, seen far more commonly in Black persons, due to selective genetic pressure in hypermalarious zones, although this trait exacts the terrible toll of sickle cell disease in homozygote sickle cell mutation-carriers [24]. Another example with malaria is that red blood cells that lack the Duffy antigens, a glycoprotein receptor for the malaria parasite, are relatively resistant to invasion by *Plasmodium vivax* [25]. Since Black people are less likely to harbor Duffy red blood cell antigens, they will be infected far less efficiently, given comparable exposure, than most White people for this one malaria species. There are other examples of genetic selection and increased or decreased risk, but social determinants of disease still dominate overwhelmingly in predicting risk for both infectious and non-communicable diseases, and care must be taken not to conflate precise shared geographic ancestry with much less precise markers of socially constructed race. After all, a minority of Blacks harbor sickle cell traits, and malaria remains a major source of morbidity and mortality in Africa. Social determinants of disease, as with the interaction of nutrition and infection or of housing stock and insect vector exposures, were substantial discriminators between indigenous populations and their colonial occupiers [26].

Yet it is not only hindsight that illuminates these alternative explanations. Contemporaneous figures put forth theories and experimental results that rendered wholly racial explanations of infectious disease incomplete at best. Germ theory discourse offered potential alternative explanations for infectious diseases, and early work in social medicine and cholera by John Snow, for example, lent credence to the impact of social determinants of disease. John Snow demonstrated that homes in more affluent parts of London were less likely to have water that was contaminated with cholera compared to those in poorer neighborhoods [27]. Kenneth MacLeod of the Indian Medical Service said “while in India natives appear to be readily susceptible to the plague, Europeans, though not absolutely insusceptible, exhibit a comparative immunity... This immunity is doubtless a sanitary immunity due to purer personal, domestic and social life, and perhaps to circumstances and habits rendering admission of infection less easy” [28] (p. 296). The language that MacLeod used still characterized White populations as “purer” and therefore superior, yet it also acknowledged the confluence of factors that contributed to varied rates of disease between colonial subjects and their White colonizers. The role of race in predicting disease was contested by more observant investigators, even in colonial times, despite the presence of race-based explanations of disease in prominent medical texts. 

Indigenous people were not only cast as uniquely immune to colonial maladies, they were also described as disease reservoirs, threatening the health of colonial White people. Henry Lippincott, chief surgeon during the American Philippine Expedition, wrote: “The Filipinos are never free from contagious diseases of one form or another, and we can never be sure they are not bringing infection into our midst.” [17] (p. 110). Victor G. Heiser, a leprologist, claimed: “As long as the Oriental was allowed to remain disease-ridden, he was a constant threat to the Occidental who clung to the idea that he could keep himself healthy in a small, disease-ringed circle” [17] (p. 112). Caring for colonized persons to protect their colonizers may seem archaic and racist, but it was also a strategy for well-intentioned doctors to coax colonial administrators to fund public health and medical care for indigenous persons. Contemporary public health advocates made a similar case that global health was in the self-interest of the United States in 2017, in an effort to discourage the hemorrhage of overseas aid and research support threatened by the Trump Administration [29]. Thus, selected positive legacies of tropical medicine in contemporary global health were co-opted to advocate for health resources for formerly colonized persons; the inherent morality of seeking “health for all” is complemented by arguments that good health for persons in LMICs also serves the vital and strategic interests of high-income nations [30].

The ongoing pandemic of coronavirus disease 2019 (COVID-19) has also exposed controversies that reinforce a perception of Western parachute researchers’ disrespect for non-Western populations. As of February 2021, COVID-19 had infected 12 times and killed 19 times more people per million population in France than in the WHO African Region (acknowledging that surveillance is better developed in France) [31]. However, during a televised interview in mid-April 2020, Jean-Paul Mira, head of the intensive care unit at the Cochin Hospital in Paris, and Camille Locht, research director at France’s National Institute of Health and Medical Research (INSERM), suggested that African populations would be ideal for the first COVID-19 clinical trials because their over-exposed and under-protected bodies, and diminished agency, could be manipulated to researchers’ advantage, without consideration of the ethics of preventable risks and passive acceptance. “If I can be provocative”, Dr. Mira argued, “shouldn’t we be conducting this study in Africa where there are no masks, no treatments, no intensive care? A little bit as is done in some AIDS studies or with prostitutes? We try things because we know they are high-risk and do not use protection. What do you think?” To which Dr. Locht replied: “Well, you are right. We have been thinking, incidentally, about carrying out such a study in Africa using BCG as a placebo” [32]. 

These insensitive comments were condemned as racist [33,34,35,36], and denounced as a vestige from the “colonial mentality” by the WHO Director-General Tedros Ghebreyesus [37]. However, these words were spoken in 2020 by two well-respected medical researchers whose research has sought to advance health for all; these unfortunate comments were sobering evidence of the continued impact of unconscious racism in global health. Scientific progress in and of itself does not eliminate colonial mindsets in contemporary global health. The misstatements of Drs. Mira and Locht (men devoted to the well-being of their constituents) follow directly from the legacy of tropical medicine. Why not experiment on persons of color if they are at higher risk of disease, if they are more susceptible, and/or if the referent diseases are a threat to the health of White people, too? 

Paternalistic patterns in research continue in contemporary global health at many levels. Research on patterns of authorship in global health demonstrates that the majority of research published about LMICs is still authored by investigators from higher-income countries [38,39,40]. Meanwhile, most health journals from LMICs are constrained to low-impact factors as assessed by citation indices [41]. A related dynamic that impedes knowledge generation and accelerates institutional memory loss in LMICs is encapsulated in the concept of “epistemicide”, the erasure of alternative, non-Eurocentric, modes of knowing. In *Epistemologies of the South: Justice against Epistemicide* [42], Boaventura de Sousa Santos argues for “global cognitive justice”, the rehabilitation of knowledge and wisdom extant in the global South that Western modernity has profoundly devalued and marginalized. 

## 3. Issues in Race and Global Health Research: Some Examples

### 3.1. Gonorrhea and Syphilis Transmission Studies in Guatemala

Less well-known than the infamous US Public Health Service’s (USPHS) natural history study of untreated syphilis among poor, Black men in Alabama is a similarly unethical international study, also conducted by the USPHS, which studied the transmission of syphilis and gonorrhea from 1946 to 1948 in Guatemala [43,44,45]. The full story was not unearthed until 62 years later, in 2010, by historian Susan Reverby, who had also written about the USPHS’s study of untreated syphilis at Tuskegee [46,47,48]. 

Her discovery led to an in-depth investigation by the US Presidential Commission for the Study of Bioethical Issues and Guatemalan Ministry of Health, which published a comprehensive historical and ethical analysis in 2011 in both English and Spanish [49]. Briefly, the archives of the principal investigator, John Cutler of the USPHS, revealed a “lost” clinical research study of “natural transmission” of syphilis and gonorrhea, and the potential for penicillin prophylaxis, which Cutler’s team had conducted in Guatemala with prisoners, soldiers, and infected female sex workers. When the natural transmission studies failed to produce more than a few cases, the team began “artificial inoculation” procedures, introducing infectious material into both men and women with swabs. The inoculation phase included prisoners, soldiers, sex workers, and adult and adolescent patients in mental institutions. There was little indication that the research subjects, particularly the women sex workers, were treated when infection did occur, and due to both their scientific failure and ethical abhorrence, these studies were never published.

While seen as monstrous today, the abuse of the Guatemalan subjects reflects the more overt bias against persons of color in international medical research in the 20th Century. As Professor Reverby’s work has shown, the fact that the USPHS deemed the harm perpetrated in Guatemala to be ethically acceptable, even as the world condemned Nazi medical atrocities, was surely influenced by the fact that these were non-White foreigners. The Guatemalans were made vulnerable to exploitation due to their ethno-racial identity and global position for the production of knowledge that would benefit the West [49,50,51,52]. 

### 3.2. Highest Mortality among Black South African Miners

Epidemiologic research documents that South African miners experience increased mortality compared to the general population [53]. They suffer from a syndemic of silicosis—a lung disease linked to the inhalation of high concentrations of the silica found in mines—with HIV infection and tuberculosis [54,55]. Mortality has decreased slightly in this population in light of more widely available antiretroviral treatment, but racial disparities in health status persist [56]. Even former miners suffer an increased risk of mortality, often highest in the first year after cessation of mining [53,57]. Black miners have a mortality risk that is more than three times that of White miners after adjusting for other variables, including specific occupational category, commodity mined, length of time in employment, and age at exit from the mining industry. Differences between Black and White miners’ housing conditions, income, nutrition, job description, silica and tuberculosis exposure, and the likelihood of family living in close proximity to the mines, likely underlie these disparities [53]. 

Epidemiologic research has unearthed the persistent health effects of apartheid, a system of land dispossession and state-sponsored discrimination that established a racial hierarchy in South Africa that sequestered resources for White people and placed indigenous, Black Africans at the bottom of the social order. Post-apartheid racial gaps persist in contemporary South Africa in educational, employment, housing, nutritional, healthcare, and wealth indicators, among others [58,59]. Black miners face added hurdles upon leaving mining. They are less likely to have access to high-quality healthcare once they are no longer eligible for employment-linked health service. They also return to home communities with disproportionately high incidences of accidental injury and violence [53]. Given the inequitable distribution of wealth and opportunity along racial lines in post-apartheid South Africa, structural racism confers disproportionate mortality risks to current and former Black miners, exacerbating the ill effects of their occupational exposures. In this context, Black miners are a vulnerable population for research, making it imperative that efforts are made to ensure their fully informed consent in research in this highly researched setting [60,61,62,63].

### 3.3. Plantation Medicine on the Firestone Rubber Operation in West Africa

The West African nation of Liberia, Africa’s oldest republic, is home to the world’s largest single rubber plantation [64]. For nearly a century, it had been operated by the then-American-owned Firestone Plantations Company, to supply Americans with a reliable source of rubber, a resource of urgent strategic import during the looming World War II [65]. At the time, Liberia was the lone source of concentrated natural latex for the Allied forces, and by 1949, it supplied nearly one-fourth (22%) of all concentrated natural latex imported to the US [66]. As detailed by medical historian Gregg Mitman in *Empire of Rubber: Firestone’s Scramble for Land and Power in Liberia* [66], the introduction in Liberia of the prized rubber tree, *Hevea brasiliensis*, and the subsequent engineering of high-yielding clones that would triple the amount of latex harvested from the trees in the war years, was a scientific tour de force that required significant investments in medical services, as well as in research in life and biomedical sciences to keep parasites and pathogens at bay. 

At its inception in 1926, the Firestone operation in Liberia was touted by its founder as “America’s greatest investment in the tropics” [66] (p. 70), that will soon stamp out malaria entirely and pioneer the development of indigenous inhabitants by bringing “civilization to darkest Africa [through] doctors, sanitary workers, civil and mechanical engineers, architects, builders, foresters, and soil experts” [66] (p. 70). The prominent African American social scientist, WEB Du Bois, an early supporter of Firestone’s beguiling vision, envisioned mobilizing a “Talented Tenth” of “colored American citizens” to help lead Liberia’s development [67]. However, Firestone instead enlisted the expertise of elite academic institutions and disciplines that enabled and benefitted from plantation indentured services [66], betraying both the true color and ultimate goal of its secular mission: white capital accumulation by local dispossession. Physical anthropologists, missionaries, and tropical medicine specialists were consulted to determine which anthropometric characteristics of which Liberian tribe would be most suited for plantation work [66]. Men from the *Kpelle* tribe, described by the team leader of the Firestone-sponsored Harvard African Expedition as “peaceful, industrious, [and formerly], principal purveyors of slaves” [66] (p. 97), stood out as the best fit, and thus becoming the largest ethnic group working as tappers in the Firestone plantation complex. 

Tappers, the most essential workers in the chain of extraction, were also the most exposed and disposable. Firestone protected its trees with more care than it treated its Liberian tappers who lived and worked in an environment rife with biological and chemical hazards [66]. To protect rubber trees from brown root rot and fungus-inflicted black thread, two major phytopathologies prevalent in the plantations, 2,4,5-T, a dioxin-containing herbicide, and Captafol, a carcinogenic fungicide, were used extensively throughout Firestone’s plantations, exposing both the workers and the surrounding communities living downstream along the Du and Farmington rivers [66]. Likewise, ammonia, a powerful base used daily by tappers to keep latex, the prized sap tapped from the bark of the rubber tree, in a liquid state was handled without gloves or protective eye gear [66]. Exposure to high concentrations of ammonia may cause immediate burning of the eyes, nose, throat and respiratory tract, and can result in blindness, lung damage or even death [68]. Some tappers lost their sight when the toxic chemical got into their eyes [66].

Lawsuits brought against Firestone in the war years by injured Liberian workers or bereaved families accused the company’s foreign doctors of “gross negligence” and “resorting to unwholesome medical experiments on their African patients”, while “in search of tropical medical knowledge and experience unobtainable in the Americas and Europe” [66] (p. 208). Political pressure from the Liberian government ultimately compelled Firestone to establish a worker compensation scheme in 1949. The surviving family of a tapper who died from work-related causes in 1950 was entitled to USD 252 representing 1400 times his daily wage (USD 0.18) [66]. The injured tapper could receive USD 115.20 (USD 0.18 × 640) for losing his eyesight and USD 120 (USD 0.18 × 700) for losing the use of either a hand or a foot [66]. As Mitman aptly noted, this was a “valuation founded upon structures of racial capitalism rooted in the violent and bloody soil of plantation slavery” [66] (p. 208).

In public relation campaign films celebrating its medical philanthropy [69,70], Firestone touted itself for providing its Liberian patients with high-standard care comparable to “that offered in hospitals in Des Moines, Hartford, Tucson, Portland, New Orleans, or San Diego” [69]. Yet, in its two-story Harbel plantation hospital, Firestone operated a segregated healthcare system that excluded even members of the Liberian political elite from receiving emergency care in the much better upstairs section of the hospital, which, just like its exclusive social club, the Firestone Overseas Club, catered to Whites only [66]. Prophylactic treatments and malaria studies conducted in residential camps on the Firestone rubber plantation [71], and later on at the nearby Firestone-sponsored Liberian Institute of the American Foundation for Tropical Medicine [72], suggest that Liberian families living on the Firestone concession were perceived as mere reservoirs of tropical disease. Therefore, they could be over-tested, over-drugged and intentionally infected with new genera of parasites, such as *P. vivax*, introduced in Liberia from Madagascar for the purpose of research [72]. To shield its White foreign staff from *P. falciparum*, the deadliest genus of the protozoan parasite that causes malaria in humans [73], Firestone required domestic housekeepers employed by its expatriate staff to receive a daily dose of quinine and plasmoquine (an international version of chloroquine). Plasmoquine was not recommended for routine use by the US Army Office of the Surgeon General [66]. Its White foreign staff, perceived as noncontagious or at least less threatening in the racial logics that informed plantation prophylaxis, were prescribed a three-times-a-week regimen of atabrine (as known in Liberia; better known as mepacrine or quinacrine) [66].

### 3.4. Marginalization of Field Scientists from LMICs: Jean-Jacques Muyembe and Ebola Virus

On March 12, 1977, *The Lancet* published three separate but related papers by teams based in Antwerp, Belgium (Institute of Tropical Medicine), Porton Down, England (Microbiological Research Establishment), and Atlanta, Georgia, USA (Center for Disease Control, its name at that time) [74,75,76]. The papers presented research on the etiology of a highly fatal hemorrhagic fever in Zaire, now named the Democratic Republic of the Congo (DRC). The newly isolated causal agent of the hemorrhagic fever was named *Ebola* virus after the Ebola river, located north of Yambuku that was the “village of origin of the patient from whom the first isolate was obtained” [76]. Thus began the incomplete story of a scientific breakthrough that excluded local, African contribution and recognized only Western scientists. 

Of note, the patient from whom the first *Ebola* isolate was obtained was Fleming (i.e., native of Flanders in northern Belgium) [77]. Although she was serving as a nun in Yambuku at the time of the outbreak, this was not her village of origin. More importantly, the first physician–scientist to observe this Flemish nun was Professor Jean-Jacques Muyembe-Tamfum, a prominent Congolese microbiologist, who has devoted four decades of an illustrious medical research career studying *Ebola* in his native DRC [78,79,80]. Before the arrival of any international team in Zaire during the 1976 *Ebola* epidemic, he was in the field, leading a national team of first responders to the epicenter of the epidemic in Yambuku [77,78,79]. While there, he recognized the importance of the study and the clinical care of the 42-year-old symptomatic nun (referred to as “patient M.E.” in the aforementioned *Lancet* papers) from Yambuku to Kinshasa, where her blood sample was collected and sent to a laboratory at the Institute of Tropical Medicine in Belgium for examination [79]. It is from this blood specimen that the *Ebola* virus was isolated, i.e., discovered. However, none of the earliest research publications associated with the discovery of *Ebola* included Muyembe’s name. In a 2018 interview with editorial staff of the *Bulletin of the World Health Organization*, he narrated his involvement in the 1976 *Ebola* epidemic as follows:

Back then, we used our bare hands, we had no protective clothing. Later, I took some liver samples from two corpses, using a steel rod and so of course there was even more blood, and again I washed [my hands] with soap and water. The liver biopsies were inconclusive, but then I examined a Belgian nun who had developed a fever, and I said to her, “since we don’t know how to diagnose this disease, I’m going to take you to Kinshasa”. She said: “I can’t leave because they’re going to think that I’m escaping because of the disease”. Finally, we persuaded her and we left. When we got to Kinshasa, we took a blood sample from her and sent it to the Institute of Tropical Medicine in Antwerp (Belgium) where Peter Piot worked. It was from the blood of this nun that Piot first isolated the *Ebola* virus [79] (p. 804).

A total of 15 persons were named as authors on the three *Lancet* papers [74,75,76], and thus credited with the co-discovery of the *Ebola* virus. All were White males and Western-based, except for Jacques Courteille, a Belgian physician who worked at the Clinique Ngaliema in Kinshasa and was credited with collecting the blood specimen [74,77]. A script similar to that of the discovery of the *Ebola* virus disease was also written in January 1970 when the scientific community was introduced to *Lassa* fever, a rodent-to-human transmitted viral hemorrhagic fever that was first observed in January 1969 in Lassa, a town in Northeast Nigeria, after a Western missionary nurse, the index case, became fatally ill [80]. All but 1 of the 13 authors of the four *American Journal of Tropical Medicine and Hygiene* papers that first described this disease were Western investigators affiliated with elite US academic institutions [80,81,82,83]. The lone “local” (and posthumous) author included in this scientific breakthrough was Dr. Jeannette Troup, a White American missionary doctor affiliated with a mission-operated hospital in Jos, North-Central Nigeria [80]. Dr. Peter Piot, a co-author on one of the *Lancet* papers on *Ebola*, admitted in a 2019 interview that in those days, “African scientists were simply excluded. White scientists—with a colonial mentality—parachuted in, took samples, wrote papers that were published in the West and took all of the credit” [84]. As evidenced by the awarding in 1951 of the Nobel Prize to Max Theiler, a member of the Firestone-funded 1926 Harvard African expedition in Liberia, for his work on a yellow fever vaccine, the “blood, parasites, and viruses collected on these [tropical] expeditions were the stuff of Nobel Prizes, professional prestige and fame, and medical breakthroughs that benefited people throughout the world” [85] (p. 1764). However, biomedical research did relatively little in return to help capacitate local medical investigators and institutions [85].

Recollecting his second encounter with Professor Muyembe upon his return to Zaire in 1983 at the onset of yet another newly emerging epidemic, AIDS, Dr. Piot stated in his 2012 memoir: “I knew him from Ebola days, as he has led the first team that went to Yambuku before the internationals arrived. He asked us for subsidies, scholarships, a budget to help set up a group on AIDS work, reagents, and also a commitment to produce *two* publications, so that the staff from the University of Kinshasa will also get scientific recognition” [77] (p. 134). That the Congolese professor and medical school dean felt compelled to ask for so much support from a junior peer from the West at the time of a local health crisis speaks to both the dire needs of many LMIC institutions and researchers and the power asymmetries that structure their collaborations with Western peers. The request for a commitment of “two publications” likely stemmed from the unfair exclusion of Professor Muyembe from the *Lancet* research articles on the *Ebola* virus. Reflecting on this early omission, Muyembe noted: “If you don’t recognize the work done in the field, it is not correct. It is a team [effort]” [84]. 

Today, Dr. Jean-Jacques Muyembe is well known among *Ebola* virus disease experts and has been given due credit for his important contributions through multiple international awards, including his pioneering of an efficacious monoclonal antibody treatment for *Ebola* [86]. The notoriety and multiple accolades he has received of late may reflect not merely his recognition as a prominent infectious disease scientist [87,88], but also an expression of Western guilt resulting from an early epistemicide. This marginalization of local investigators has also been observed in South Africa, where Black South African contributors have been systematically neglected for proper research credit compared to other South African investigators [89].

## 4. Current US-Led Efforts in Global Health Research

### 4.1. The Fogarty International Center Model in HIV Research

In many LMICs, medical, public health, and nursing education and training in human health research have been limited to a privileged few. International trainees of color typically have been shunted to clinical care roles, with research reserved for elites. Recognizing these workforce disparities, many donor nations have focused their international programs to highlight former colonies, now independent nations. One such example is the John E. Fogarty International Center (FIC) for Advanced Study in the Health Sciences at the National Institutes of Health (NIH), which was established in 1968 [90]. In its early years, FIC/NIH convened and hosted global scientists from high-income nations in Europe and Japan to nurture collaborations with US scientists. In the 1980s, however, the global HIV/AIDS pandemic focused the urgency of engaging LMICs and providing training opportunities for indigenous investigators who could tackle autochthonous challenges in their home nations [90,91,92,93,94,95,96]. Since 1989, more than 7500 individuals from over 130 countries have trained through FIC programs [97]. 

### 4.2. Research Leadership from LMIC Investigators of Color

One of the FIC/NIH’s priorities is to help train and capacitate investigators and institutions in LMICs. Their largest initiative to prepare international researchers, alongside US citizens and permanent residents who are focused on global health, is the FIC Global Health Scholars and Fellows Program, which has trained over 1000 aspiring scientists in one-year field experiences in global health in LMICs [98,99,100,101,102,103,104,105]. The FIC is joined by other NIH institutes in seeking to nurture investigators of color in LMICs, as with the HIV Prevention Trials Network Scholars Program, which has an international component to complement a US domestic program [106]. While beyond the scope of this paper, many other nations, both high-income and LMICs themselves, are engaged in individual research capacity-building through governmental, private sector, non-profit foundation, or other support. Still, the challenges do not stop with training per se. Ongoing research mentorship is needed for all young scientists if they are to be successful, and this support has been highlighted by additional FIC/NIH investments [107,108,109,110,111,112,113]. LMIC mentors and institutional leaders express enthusiasm for the programs [114].

### 4.3. Improving Local Research Environments in LMICs

Increasingly, FIC/NIH’s investments are helping in building programs and institutions, not merely individuals. Without strong research support and context, the training of scientists from LMICs may risk exacerbating “brain drain”, the departure of persons whose research expectations cannot be met in their home nations [115,116,117,118]. Poor infrastructure and limited career support can incentivize the emigration of clinicians and researchers alike. FIC/NIH made a large investment in research capacity within the context of the US President’s Emergency Response for AIDS Relief (PEPFAR), termed the Medical Education Partnership Initiative (MEPI) and the Nursing Education Partnership Initiative (NEPI) in sub-Saharan Africa [119,120,121,122,123,124,125,126]. FIC/NIH supports many programs to build research workforce in specific areas of biomedical and behavioral research in LMICs. Examples include environmental health and climate change [127,128], data sciences [129,130], the control and prevention of infections and pandemics [131,132,133,134,135,136], and the health of vulnerable populations and those with special conditions, including women, children, adolescents, and youth [137,138,139,140,141,142]. A number of institutions that have been developed as local and regional research centers have distinguished histories of furthering research of high community relevance. For example, the Kenya Medical Research Institute (KEMRI) has made decades-long contributions in malaria control [143,144,145], and the International Centre for Diarrhoeal Disease Research, Bangladesh (ICDDR,B) has undertaken fundamental work to develop an oral rehydration solution for infant diarrhea treatment [146,147,148].

The HIV/AIDS pandemic was a strong driver of FIC/NIH’s pivot towards capacity building in LMICs [92,119,149]. Important financial investments in HIV/AIDS research have also been credited with the rise of global health programs across top research universities in the global North. The increasing desirability of “partnerships” with institutions in sub-Saharan Africa has become intense; one critique has characterized these asymmetric partnership trends as a new “Scrambling for Africa” [1]. More recently, non-communicable diseases (NCDs) have been highlighted as posing a neglected challenge in LMICs [150,151,152,153]. FIC/NIH programs have advocated the use of the PEPFAR infrastructure that supports the chronic disease care of persons living with HIV and persons on long-term therapy and monitoring for tuberculosis to build clinical, community, and research capacity for NCDs as well [154,155,156,157,158]. FIC/NIH has also invested in strengthening health systems, developing capacity in policy, and building educational institutions in this area, often with co-funding from other sources [159,160,161,162,163,164,165,166]. 

## 5. Discussion

When unethical global health research is identified, it is often related to the same issues that have disenfranchised communities of color globally from equitable health care: limited input in the research, few to no investigators from the community, vulnerable research subjects who have not provided truly informed consent, sometimes questionable ethical review, and unfair resource allocation. Respect for persons, beneficence, and justice are lacking. Key reforms have been initiated globally to counteract the decades of exclusion of research investigators of color, community voices in research, and human rights-based ethics reviews and monitoring. 

### 5.1. Ethical Guidance for Global Health Research

Without doubt, the formalization of standards of research ethics has been central to improvements in international research with populations of color in LMICs. The 1979 *Belmont Report* referred explicitly to principles articulated by the Nuremberg Code of 1947, crafted to address Nazi research abuses of racial minority populations during World War II [167,168,169,170,171]. The three central principles articulated in the *Belmont Report* for ethical human biomedical and behavioral research are just as applicable to research in LMICs as in Europe or North America: *Respect for persons*, treating individuals as autonomous agents who must agree to research by informed choice, and protecting the interests of persons with diminished autonomy as with children or prisoners;*Beneficence*, proposing only research that has the prospect of benefiting society, ideally also benefiting, or at least not harming, the research volunteers;*Justice*, including participants in research relevant to persons who can most benefit from the work, advantaging the largest pool of individuals by the research findings, and not simply those who are most convenient to enroll.

Guidelines for international research, notably the Declaration of Helsinki and the Council for International Organizations of Medical Sciences’ (CIOMS) guidelines, also embody these principles. The 1964 World Medical Association’s *WMA Declaration of Helsinki—Ethical Principles for Medical Research Involving Human Subjects* is a 37-point document that first established international norms for human subjects research, now in its 8th edition [172]. CIOMS’ *International Ethical Guidelines for Health-related Research Involving Humans* have undergone multiple revisions since the first published version in 1982, but their aim has remained consistent: “to provide internationally vetted ethical principles and detailed commentary on how universal ethical principles should be applied, with particular attention to conducting research in low-resource settings” [173]. Of note, “low-resource settings” exist in middle- and high-income countries and should not be narrowly construed as “low-resource countries” [173]. A 2010 analysis of data from the US Bureau of Justice Statistics and the US Census Bureau suggests that the typical US environments in which many African Americans live could qualify as low-resource settings, since such environments, even after controlling for violent and accidental deaths, are often more harmful to many Black males’ lives than US prisons, where they receive better healthcare and have better health outcomes [174].

In 2013, at the Global Maternal Health Conference held in Arusha, Tanzania, an international team (Agnes Binagwaho, Wendy Graham, Rafael Lozano, and Marleen Temmerman) drafted a *Code of Conduct* specific to research in low-income countries to improve equity and opportunity for local investigators: No ethics committee, funder of research, or medical journal should approve, support, or publish research about a low-income country without joint authorship from that country;In any research project in a low-income setting, local scientists must be included as co-principal investigators;Before starting research in a low-income country, Western authors and institutions must define a clear plan for how they will transfer research skills back to that country;Medical journals and their publishers must ensure that all global health research is free at the point of use in countries;Western journals must facilitate the language translation of research, either themselves or by enabling local journals to republish freely [175] (p. 278).

Among other influential documents in this area is the 2001 report of the US National Bioethics Advisory Commission, *Ethical and Policy Issues in International Research: Clinical Trials in Developing Countries* [176]. The report contains 28 recommendations that focus on the main ethical requirements for the conduct of clinical trials overseas by US interests, and the necessity for such trials to be directly relevant to the health needs of the host population. Notably, the report recommends that the Food and Drug Administration (FDA) (or presumably the European Medicines Association, the South African Health Products Regulatory Authority, and others) should not accept data from clinical trials where lapses in ethical protections of participants are reported [176].

Modern guidance on the ethics of research with human subjects has also developed specific standards for the special circumstances in LMICs [177]. One area where the risk–benefit calculus may differ in varying circumstances is illustrated by HIV prevention research, for example [177,178]. However, ethical codes, in and of themselves, are not panaceas. The USPHS syphilis experiments in Guatemala took place from 1946 to 1948, just as Nazi abuses were coming to light and the 1947 Nuremberg Code was written, and the USPHS studies at Tuskegee extended from 1932 to 1972, ending fully 25 years after the Nuremberg Code and eight years after the publication of the *Declaration of Helsinki*. 

### 5.2. Local Ethics Review Committees and Data Safety and Monitoring Boards

Better regulation and management of research with expanded capacities of in-country research agencies are needed for LMICs to have autonomy over research conducted locally. The training and development of research ethics committees (RECs) and institutional review boards (IRBs) in LMICs has been among the goals of training activities [179,180,181,182,183,184,185]. As autochthonous research capacity in LMICs expands, so will the need to ensure reviews of local research proposals by ethics review boards, provide ethical oversight with data and safety monitoring boards, and ensure expert guidance [186,187,188,189,190,191,192,193,194,195,196,197].

There are many examples of dissonance in the ethical review of research protocols. The REACH Project of the Adolescent Medicine HIV/AIDS Research Network had 11 clinical sites with a single, common protocol, yet multiple IRBs provided different assessments of research risks and benefits [198,199,200]. Such differences of opinion can extend across IRBs, funding agencies, community advocates, and journal editors in different countries, as seen in the case of perinatal studies of antiretroviral drugs to prevent mother-to-child HIV transmission in Thailand in the late 1990s. Funders, government agencies, and both US and Thai IRBs approved protocols for placebo-controlled studies of short-course zidovudine, since higher doses that also involved intravenous administration were deemed unaffordable to replicate in low-resource settings [201,202]. The rationale was that the cost–benefit for such studies was different for countries without access to high-cost interventions [203,204,205,206]. Yet some advocates (and even the journal editor who published the studies) declared them to be unethical, since they denied access to pregnant women living with HIV to standard of care prophylactic therapies [207,208]. In publishing the work, the journal editor appeared to defer to the judgment of the Thai and US IRBs as to the ethical conduct of the study in Thailand, but protested its ethical basis in the same issue via editorial commentaries, many letters, and a reply from the authors [203,209]. Results from a subsequent, similar trial from Thailand were published in a different journal, with no objection from the responsible editor [204].

The debate over local versus international ethical standards for research in LMICs continues today. For example, if cardiomyopathy from Chagas disease is most prevalent among the poor in Brazil, is it ethical to conduct research to mitigate cardiac disease when there is a definitive global standard of care (cardiac transplant) that is unavailable or unaffordable for the persons beleaguered by the condition? Some would argue that it would be unethical to forego research to mitigate the cardiac harms for this disease, which overwhelmingly affects poor persons with thatched roof housing that is friendly to the reduviid bug vector. Others would argue that ethical standards must be universal, and that research proposing something inferior to definitive, evidence-based therapy should not be permitted. The issue arises as to whether and how the risk–benefit assessment for such decisions might differ between high- and low-incidence nations. 

### 5.3. The Role of Scientific Journals

Journals have an important role in expanding the participation of LMIC researchers in global health. Typical criteria for authorship are derived from multiple versions of similar standards that can be summarized as contributing substantially: (1) to the conceptualization and design of the study; (2) to the conduct of the study and collection or analysis of the data; and (3) to the interpretation of findings and drafting of the manuscript. Approval of the manuscript is also required for authorship. A regrettable continuing inequity is that researchers in LMICs, particularly persons of color, may not be given the opportunity to contribute to all of these roles, and therefore may be systematically underrepresented in the authorship of published manuscripts. In a concerted effort to establish equity among investigators by forcing this issue of opportunity, a few journals refuse to review and/or publish work from a study originating in an LMIC that does not feature an investigator from that nation as a co-author [16]. A bibliographic analysis of publications from 2009–2017 suggests that LMIC authors are now far better represented today than previously. However, increased authorship from Chinese investigators was a dominant contributor to the improved metrics [210]. 

With the rise and rapid growth of internet-based mega-journals [211], local journals in LMICs face immense competition, which is further compounded by the growing threat of predatory journals [212,213,214]. A 2017 study published in the *South African Journal of Science* estimated at 3.4% the extent of predatory publishing among South African authors during the 2005–2014 decade. When disaggregated by institution, publication in predatory journals among South African authors ranged from <1% among authors affiliated with the University of Cape Town, Rhodes University, and the University of the Witwatersrand, to >25% among authors affiliated with Walter Sisulu University, University of Fort Hare, and Mangosuthu University of Technology from 2005 to 2014 [215]. As emerging authors of color seek to publish in international journals to advance their careers, perverse incentives can drive knowledge production away from the region where the research was conducted, estrange local clinicians from the research process, and possibly lessen the impact of the research locally, even as it is promulgated globally. Thus, investigators must take on the responsibility to disseminate their work locally and apply their research findings in LMIC settings.

## 6. Conclusions and Future Directions

Human subjects research has expanded markedly in past decades, and more research work is performed in low- and middle-income countries (LMICs) than ever before. That the HIV pandemic was so much more severe in sub-Saharan Africa than in any other region or continent has moved much of the world’s prevention research to high-incidence communities in sub-Saharan Africa, building human subjects research infrastructures beyond anything previously developed. Emerging infections and neglected tropical diseases have also marshalled attention towards epidemiologic, behavioral, vaccine, and clinical research so as to clarify strategies for disease control and prevention. Ebola virus in Africa, Zika virus in South America and the Caribbean, and severe acute respiratory syndrome coronavirus 2 (SARS-CoV-2) in Asia are but three recent examples.

Notwithstanding notable improvements in research ethics, racial bias and discrimination are still evident in the global research arena in many areas, particularly in LMICs—for a recent example, see Erondu et al. [216]. Internationally, persons of color may remain at disproportionate risk for being entered into trials that racialize disease; that are unethically designed; that have not fully informed their participants about what the study involves; that have not fully engaged investigators from the host nation; or that have not included participants from minority groups within the venue where the study is being conducted. Racism in international research continues to threaten both the scientific integrity and ethical conduct of global health. To ensure the fair resolution of these issues in individual studies and global health research generally, RECs/ IRBs, governments, communities, and investigators must wrestle with them in a spirit of inclusivity, equity, and anti-racism.

### 6.1. Aiming for Self-Reliance and Sustainability in Global Health Research Funding

While global health no longer operates under the “tropical medicine” paradigm in which most investigators are White people from high-income nations and most research subjects are persons of color from LMICs, it remains a constant struggle to achieve even a sliver of health equity between higher- and lower-income nations, and White investigators and investigators of color. The existence and sustainability of the academic field of global health holds paradoxes; scientific opportunities are exploited for the generation of knowledge even as the disparities in disease in LMICs are recognized as preventable and contributing to avoidable suffering [1]. Since money flows largely from higher-income nations, authority and opportunity routinely follow the money. When LMIC institutions can be self-reliant with (rare) endowments or supported by a reliable extramural grant stream, equity can be nurtured. However, these channels of support are most available for already-mature global institutions where systematic efforts to bring leaders into high-impact research networks are most successful.

### 6.2. Including Minority Populations from LMICs

In nations that have dominant majority populations that have and continue to hold hegemonic power (White people in North America and Europe, Han ethnic group in China, etc.), it is incumbent upon investigators to seek to enroll participants from minority groups into clinical research studies whenever possible. In China, this may mean the recruitment of members of multiple Chinese ethnic minority groups, some of which have over 2 million members, for example: Zhuang (16.9 million), Hui (10.5 million), Manchu (10.3 million), Uyghur (10 million), Miao (9.4 million), Yi (8.7 million), Tujia (8.3 million), Tibetan (6.2 million), Mongol (5.9 million), Dong (2.8 million), Buyei (2.8 million), Yao (2.7 million), and Bai (1.9 million) [217]. In India, this may imply a greater representation of minorities with intersecting identities, for example Dalit Christians and Muslims, or religious minorities (e.g., Sikhs, Buddhists, Jains) who are also linguistic minorities (e.g., Gujarati, Telugu, Tamil, Urdu) or belong to tribal and indigenous communities, like the Boros, Adivasis, and Andaman Islanders [218]. In South Africa, 13 ethnic groups speak the 11 official languages of that nation: Pedi, Sotho, Tswana, Swati, Venḓa, Tsonga, Afrikaans, English, Ndebele, Xhosa, and Zulu [219]. A survey of group self-identity conducted in South Africa found considerable heterogeneity and complexity in intergroup relations [220]. 

The fact that different ethnic groups live in different provinces of China, India, or South Africa suggests that not all ethnic groups can join a given venue-based research study. Still, an effort to maximize opportunities for participation and improve the generalizability of study results is an ethical imperative. As in high-income nations, members of minority groups in LMICs must be considered for inclusion in research, particularly in research from which they may benefit as a group. Moreover, while the focus of this article is *race* and research, other marginalized structural identities that intersect with a compounding oppressive effect also warrant attention, including minority status by sex and/or gender identity, age, sexual orientation, socioeconomic status, caste, rural or urban residence, housing status, and disability status.

## Figures and Tables

**Table 1 ijerph-20-06210-t001:** PubMed search results for Africa, China, India, and the United States.

Query	Total Citations (Until December 2022)	2012–2022 “Decade”	Percent Change
Africa NOT (China OR India OR United States)	377,022	189,515	50.26%
India NOT (Africa OR China OR United States)	690,238	461,067	66.79%
China NOT (Africa OR India OR United States)	2,416,754	1,969,804	81.50%
United States NOT (Africa OR China OR India)	4,182,876	1,721,209	41.14%

## Data Availability

Data sharing is not applicable to this review article.

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
