# Peer review of "Global Health Perspectives on Race in Research: Neocolonial Extraction and Local Marginalization"

_ijerph, 2023, doi:10.3390/ijerph20136210_

Round 1
Reviewer 1 Report
This is a well written and well thought review article which I thought would contribute positively to the field of population/global health, and as well as laying a good foundation for future large scale research study. Haven said that, I would like the authors to address the fees concerns listed below;
1. Lines 26-35: it looks more like a "material and methods" section of the article than introduction in my opinion. I will suggest the authors re-writes those lines and attach the data described in the above mentioned lines as a supplementary document.
2. Page 360: I will suggest the authors include the story on the origin of Lassa fever; caused by Lassa fever virus which is a hemorrhagic fever named after a town called Lassa in Bornu state, Nigeria. will be interesting to mention that while on the topic of hemorrhagic fever virus origin. Opara, N.U.; Nwagbara, U.I.; Hlongwana, K.W. The COVID-19 Impact on the Trends in Yellow Fever and Lassa Fever Infections in Nigeria. Infect. Dis. Rep. 2022, 14, 932-941. https://doi.org/10.3390/idr14060091
Author Response
1. Lines 26-35: it looks more like a "material and methods" section of the article than introduction in my opinion. I will suggest the authors re-writes those lines and attach the data described in the above mentioned lines as a supplementary document.
Response: We have moved the data reported in the first paragraph of the Introduction into a table (lines 37-38), after revising the paragraph as suggested by the reviewer. However, we did not see the necessity or the appropriateness of saving the information as supplemental material, as this will most likely conceal the information and will ensure that most readers miss it.
2. Page 360: I will suggest the authors include the story on the origin of Lassa fever; caused by Lassa fever virus which is a hemorrhagic fever named after a town called Lassa in Bornu state, Nigeria. will be interesting to mention that while on the topic of hemorrhagic fever virus origin. Opara, N.U.; Nwagbara, U.I.; Hlongwana, K.W. The COVID-19 Impact on the Trends in Yellow Fever and Lassa Fever Infections in Nigeria. Infect. Dis. Rep. 2022, 14, 932-941. https://doi.org/10.3390/idr14060091
Response: Per the reviewer's suggestion, we have included the story of the Lassa fever discovery (additional new material not included in the book chapter). This new information can be found in lines 389-398. However, we have supported our statement with references that are most relevant to the discovery story of the Lassa fever than the one suggested by the reviewer.
Reviewer 2 Report
I found this manuscript fascinating. It combines Washington's MEDICAL APARTHEID with Fanon's WRETCHED OF THE EARTH. It is incredibly researched and highly analytical, devoid of emotional pitying but loaded with compassion and can move even the most conservative. I read and re-read it, and could not find anything of substance to complain or offer helpful feedback. I was an IRB chair at my university at one time and can readily attest to the attitude of the research enterprise, especially health research, bending to the will of the academic status-quo to the detriment the people or participants, or subjects, or what is actually research fodder. The content of the manuscript itself could serve, in my view, as the guideline for an entire course by itself that is desperately in global health. There are enough substantive references for student to pursue further reading if they choose, or as stated, the manuscript can stand alone as a complete syllabus.
Author Response
Thank you very much for the positive impression.